# Application of Next-Generation Sequencing (NGS) Techniques for Selected Companion Animals

**DOI:** 10.3390/ani14111578

**Published:** 2024-05-27

**Authors:** Kinga Domrazek, Piotr Jurka

**Affiliations:** Institute of Veterinary Medicine, Warsaw University of Life Sciences—SGGW, Nowoursynowska 159c, 02-776 Warsaw, Poland; piotr_jurka@sggw.edu.pl

**Keywords:** NGS, companion animals, diagnostic tools

## Abstract

**Simple Summary:**

This review explores the wide-ranging applications of Next-Generation Sequencing (NGS) techniques in veterinary medicine for companion animals. It addresses the need for advanced diagnostic and treatment approaches in veterinary care. The aims include understanding parasitic transmission, infectious disease dynamics, metabolic disorders, autoimmune diseases, oncological processes, and genetic predispositions in animals. The article demonstrates NGS’s efficacy in comprehensive parasite profiling, rapid pathogen identification, personalized tumor analysis, and genetic disease elucidation. Conclusions highlight the transformative potential of NGS in enhancing veterinary diagnostics, treatment planning, and disease management for companion animals. The review’s insights promise to revolutionize veterinary care, enabling precise and tailored treatments, improving animal health outcomes, and advancing overall societal well-being through healthier and happier companion animals.

**Abstract:**

Next-Generation Sequencing (NGS) techniques have revolutionized veterinary medicine for cats and dogs, offering insights across various domains. In veterinary parasitology, NGS enables comprehensive profiling of parasite populations, aiding in understanding transmission dynamics and drug resistance mechanisms. In infectious diseases, NGS facilitates rapid pathogen identification, characterization of virulence factors, and tracking of outbreaks. Moreover, NGS sheds light on metabolic processes by elucidating gene expression patterns and metabolic pathways, essential for diagnosing metabolic disorders and designing tailored treatments. In autoimmune diseases, NGS helps identify genetic predispositions and molecular mechanisms underlying immune dysregulation. Veterinary oncology benefits from NGS through personalized tumor profiling, mutation analysis, and identification of therapeutic targets, fostering precision medicine approaches. Additionally, NGS plays a pivotal role in veterinary genetics, unraveling the genetic basis of inherited diseases and facilitating breeding programs for healthier animals. Physiological investigations leverage NGS to explore complex biological systems, unraveling gene–environment interactions and molecular pathways governing health and disease. Application of NGS in treatment planning enhances precision and efficacy by enabling personalized therapeutic strategies tailored to individual animals and their diseases, ultimately advancing veterinary care for companion animals.

## 1. Introduction

In recent years, the landscape of molecular diagnostics in veterinary practice has been profoundly influenced by advancements in molecular biology and genetics [1]. In the preceding years, a significant amount of research has been based on the polymerase chain reaction (PCR) and its variations [2]. Real-time PCR (RT-PCR) is still used frequently in everyday veterinary practice, as clinicians have access to the technology, and the test is relatively inexpensive. 

The second big step in developing molecular diagnostic was first-generation sequencing. Sanger introduced his “plus and minus” method for DNA sequencing in the 1970s. This pioneering technique marked a crucial transition, paving the way for the subsequent development of sequencing methods that have since become the dominant approach in the field for the past three decades [3]. One primary advantage of this method is its capability for sequencing, wherein the result is obtained concurrently with the synthesis of the new DNA strand. This approach boasts rapidity, because of stems from its automated loading of fluorescent ddNTPs and the integration of interconnected system components such as personal computers [4]. Pyrosequencing, emerging as a result of these advancements, laid the groundwork for a novel sequencing paradigm later dubbed Next-Generation Sequencing (NGS), which is characterized by two related terms: massively parallel (simultaneously in the same manner) and deep sequencing [5].

Nowadays, with the development of molecular techniques, NGS is gaining popularity and forms the basis of many studies. This technology can sequence hundreds and thousands of genes or conduct whole-genome sequencing in a short time [6]. NGS encompasses a suite of methodologies tailored for high-throughput sequence analysis of DNA and RNA. Diverging from traditional Sanger sequencing, which demands a unique DNA template, NGS facilitates the examination of heterogeneous samples while concurrently furnishing sequence data for over 10 million nucleic acid molecules within a sample [7]. Due to its vast read capacity, NGS stands as an ideal tool for nucleic acid analysis and diagnostics necessitating multiplexed scrutiny of numerous genes and their variants [8].

The majority of current NGS technologies operate on a fundamental principle: sequencing a multitude of DNA fragments (ranging from thousands to millions) simultaneously in a single machine run. The types of sequencing technologies with advantages and disadvantages is presented in Table 1. The procedure of NGS is presented in Figure 1. This necessitates the conversion of nucleic acids (such as total DNA, genomic DNA, RNA, etc.) into machine-readable fragments through a process known as library preparation, following their extraction and purification [9]. Subsequently, the substantial amount of sequence data generated must undergo analysis using bioinformatics procedures tailored to extract the desired information for various applications [10]. The approach to library preparation and the selection of sequencing instruments and associated technologies significantly influences the potential downstream analyses [7]. 

In human medicine, diagnostic tools often undergo earlier adoption and integration into clinical practice compared to their counterparts in veterinary medicine [15]. This discrepancy can be attributed to various factors, including regulatory requirements, funding availability, and emphasis on public health concerns. The rapid evolution of diagnostic techniques in human medicine frequently serves as a catalyst for advancements in veterinary diagnostics. Techniques developed for human medical applications, such as NGS sequencing [16], imaging modalities [17], and point-of-care testing [18], often find parallel utility in veterinary medicine. Furthermore, advancements in understanding human diseases and their diagnostic markers frequently translate to similar investigations in animal health, fostering cross-disciplinary collaborations and knowledge exchange between human and veterinary medical communities [19,20]. Consequently, the development and refinement of diagnostic techniques in human medicine play a pivotal role in shaping the landscape of veterinary diagnostics, facilitating the early detection, diagnosis, and management of diseases in animals.

In recent years, NGS has also emerged as a transformative tool in veterinary medicine, revolutionizing our understanding of animal health and disease [21,22]. By enabling rapid and comprehensive analysis of DNA and RNA from diverse animal species, NGS holds immense potential for a wide range of applications in veterinary practice. From identifying genetic predispositions to diseases [22] and assessing microbial diversity in animal populations [20,23,24] to monitoring the spread of infectious agents and improving breeding programs, NGS offers unprecedented insights that are reshaping the landscape of veterinary diagnostics and research [25,26,27]. 

## 2. Physiological Investigations

Exploring physiological processes in animals is fundamental to understanding their biology, health, and adaptation to changing environments [28]. NGS has emerged as a powerful tool in physiological investigations, offering unprecedented opportunities to dissect complex molecular mechanisms underlying physiological functions. By enabling high-throughput sequencing of DNA, RNA, and epigenetic modifications, NGS has revolutionized our ability to explore gene expression, regulatory networks, and functional genomics in various animal species [29]. In their study, Nowak et al. sought to deepen understanding of the mechanisms underlying the termination of canine pregnancy by employing transcriptome sequencing [30], which involves the high-throughput sequencing of RNA molecules to quantify gene expression levels and identify transcript isoforms [31]. They analyzed placental transcriptomes derived from natural prepartum and antigestagen-induced abortions and compared them with fully developed mid-gestation placentas.

### 2.1. Reproduction Research

The comparison between “prepartum luteolysis over mid-gestation” revealed differentially expressed genes (DEGs), with overrepresented terms associated with apoptosis, impairment of vascular function, and activation of cytokine signaling pathways during natural luteolysis. In comparison, antigestagen treatment yielded highly regulated DEGs, reflecting induced luteolysis with similar functional term associations and expression patterns as natural luteolysis [30]. This study shows that NGS can find applications not only in clinical sciences but also in preclinical sciences like physiology. However, understanding the physiological processes occurring in the body can be directly applicable to clinical work with patients [4].

### 2.2. Microbiology

Since NGS excels in microbiology, it can be used to learn about physiologically resident microorganisms. Harvey et al. have evaluated the quantification of the bacterial flora and its major constituents on the abdominal skin of clinically healthy dogs [32] by using NGS technology. Interestingly, research utilizing NGS techniques unveiled a greater diversity and richness in the bacterial microbiota compared to previous studies reliant on culture-based methods. Such results were obtained by Bergeron et al. while conducting research on the microflora of anal sacks [33]. Banks et al. conducted a study evaluating the conjunctiva of healthy dogs using NGS and found a rich and diverse microbiota. Importantly, this study highlighted that traditional culture methods significantly underestimated the number of bacterial taxa present on the healthy canine ocular surface [34]. Comparison of the skin and ear canal microbiota of non-allergic and allergic German shepherd dogs using NGS have been also performed [35]. Analysis of bacterial 16S rRNA gene amplicon sequences revealed consistent bacterial community patterns across various body sites in non-allergic dogs. Notably, the microbiota composition at different body sites in non-allergic German Shepherd Dogs (GSDs) exhibited no significant disparities. However, in allergic dogs, differences emerged particularly in axilla samples, where a lower species richness was observed compared to non-allergic counterparts. Furthermore, allergic dogs displayed site-specific variations in bacterial community composition, notably in the skin and ear canals. Actinobacteria predominated in non-allergic dogs, contrasting with *Proteobacteria* dominance in allergic dogs. *Macrococcus* spp. were more prevalent on non-allergic skin, whereas *Sphingomonas* spp. were more abundant on allergic skin [35]. A similar study, conducted to evaluate the microbiome of healthy and allergic cats, has been performed by using NGS [36]. Those studies show that utilizing traditional microbiological methods to explore the microbiota presents several limitations, which can be effectively addressed through molecular procedures.

## 3. Metabolic Processes in Animals

Metabolic processes govern the intricate balance of energy production, nutrient utilization, and waste elimination in animals [37]. Understanding these processes is crucial not only for advancing animal physiology research but also for improving livestock production, enhancing animal health, and even addressing human metabolic disorders. While traditional biochemical assays and genetic studies have provided valuable insights into metabolism [38], they often offer only snapshots of the complex interplay between genes, proteins, and metabolites. Enter NGS, a powerful tool that has revolutionized our ability to comprehensively explore the molecular underpinnings of metabolic pathways in animals [39]. 

Obesity stands as a widespread health issue affecting both humans and companion animals worldwide. Its ramifications include premature mortality, metabolic irregularities, and various health complications spanning different species. Consequently, obesity holds significant relevance in both medical and veterinary domains [40]. Grzemski et al. investigated by using NGS whether the genes associated with obesity in humans play a role in canine obesity. They suggested that FTO and IRX3 are not useful markers of obesity in Labrador dogs [41]. In addition, obesity stands as a primary contributor to chronic kidney disease, yet the precise molecular pathways triggering kidney injury and dysfunction in obesity-related nephropathy remain elusive. Eritja et al. aimed to elucidate the kidney microRNA (miRNA) expression profile in a model of obesity-induced kidney disease in C57BL/6J mice, employing NGS analysis [42]. High-fat-diet-induced obesity induced marked structural changes in the tubular and glomerular regions of the kidney, alongside increased renal expression of proinflammatory and profibrotic genes, as well as elevated expression of genes involved in cellular lipid metabolism. MiRNA sequencing revealed a panel of nine differentially expressed miRNAs in the kidney following high-fat-diet feeding [42]. Rodney et al. evaluated new gene candidates, notably detecting novel missense variants associated with polycystic kidney disease in the Peterbald cat breed. These findings underscore the efficacy of Whole-Exome Sequencing (WES) in pinpointing previously unknown gene candidate alleles linked to diseases and traits, marking a significant advancement in feline model research [43].

NGS can also be used to assess cellular energy metabolism. Loria et al. performed gene expression profiling of endomyocardial biopsies from dogs with dilated cardiomyopathy phenotype [44]. In their evaluation, they found distinct gene expression pathways associated with cellular energy metabolism, particularly those involving carbohydrates, insulin, and cardiac structural proteins in all DCM-p dogs as opposed to the control group. Furthermore, when comparing dogs with lymphocytic interstitial infiltrates to the control group, NGS analysis highlighted the significant involvement of genes related to inflammation and pathogen infection [44]. 

Drug-metabolizing enzymes play a crucial role in drug development and therapy, yet their comprehensive identification and characterization across various species, lines, and breeds remain incomplete. In a study by Uno et al., liver transcriptomic data were scrutinized to assess phase I cytochromes P450, flavin-containing monooxygenases, and carboxylesterases, along with phase II UDP-glucuronosyltransferases, sulfotransferases, and glutathione S-transferases [45]. Comparative analysis across a spectrum of species including humans, rhesus macaques, African green monkeys, baboons, common marmosets, cattle, sheep, pigs, cats, dogs, rabbits, tree shrews, rats, mice, and chickens unveiled both broad similarities and discrepancies in the transcript abundances of drug-metabolizing enzymes. Furthermore, Beagle and Shiba dogs were subjected to NGS to further explore this aspect [45]. This diagnostic tool can be used in the future to develop new drugs and learn more about their metabolism [39].

## 4. Veterinary Genetic

Advancements in veterinary genetics have transformed our understanding of hereditary diseases and population dynamics in animals [40,46,47]. The integration of NGS technologies has been instrumental in accelerating progress in this field. NGS enables comprehensive analysis of the entire genome, providing unprecedented insights into genetic variation, gene expression patterns, and molecular pathways underlying various diseases in animals [10,30,44]. NGS could also be used in prenatal metabolic studies. A study performed on canine embryonic fibroblasts shows that this diagnostic tool can be used in the detection of homozygous point mutations [48].

In the study of human biology, complete sequencing of mitochondrial DNA (mtDNA) has found extensive application across various disciplines such as medicine, forensics, and genetics [49]. Similar situations could be observed in veterinary medicine. Few authors performed whole-feline-genome sequencing [50,51,52,53], which involves the determination of the complete DNA sequence of an organism’s genome, including all its genes and non-coding regions [53] Sugasawa et al. established a sequencing method for the whole mitochondrial DNA of domestic dogs [27]. Other authors have evaluated whole-genome sequencing of Chinese crested dogs [26]. Furthermore, they have pinpointed novel genetic variations within the Chinese Crested dog breed, which will enhance ongoing whole-genome sequencing investigations aimed at uncovering mutations linked to monogenic diseases with autosomal recessive inheritance [26].

Gene sequencing can also be useful for establishing paternity in dogs and cats [54]. In recent years, more and more breeders are choosing to mate one bitch or cat with two sires. This is performed to obtain puppies simultaneously from two matings, especially if it is to be the last litter of a female [55]. Such procedures are aimed at increasing the gene pool in breeds with high inbreeding rates and increasing the likelihood of producing outstanding offspring. Azirmedi et al. performed analysis of Doberman Pinscher and Toy Poodle samples with targeted next-generation sequencing by using a panel which included 383 targets [25]. The authors suggested that this method presents an economical, swift, and scalable genotyping strategy for examining parentage, assessing the occurrence of genetic disorders and traits across numerous samples, and probing genetic interplays among previously identified variants [25].

Several well-established genome projects have significantly advanced our understanding of canine and feline genetics, providing invaluable resources for both veterinary and comparative genomic research. The Canine Genome Project stands as one of the pioneering efforts in this domain [56]. Led by an international consortium of researchers, this project aimed to sequence and annotate the entire canine genome, resulting in the publication of the first draft assembly in 2005 [57]. Since then, continual improvements in sequencing technologies and bioinformatics tools have facilitated the refinement of the dog genome assembly, enabling comprehensive analyses of genetic variation, breed diversity, and disease susceptibility across canine populations. Similarly, the Feline Genome Project [58], launched with the goal of elucidating the genetic makeup of domestic cats, has made significant strides in characterizing the feline genome. Through collaborative efforts and advancements in sequencing methodologies, researchers have produced high-quality genome assemblies for various cat breeds, laying the foundation for investigating the genetic basis of feline traits, diseases, and evolutionary adaptations. These genome projects serve as invaluable resources for deciphering the genetic underpinnings of complex traits, facilitating the development of diagnostic tools, and informing breeding practices aimed at promoting animal health and welfare [57,58].

Many diseases have a genetic basis and are inherited [59]. NGS may find application first in the scientific realm in identifying these genes, and then in clinical practice during the diagnosis of patients or carriers. It will facilitate the selection of potential parents during mating planning and the diagnosis of an already affected individual. One example of genetic diseases is copper toxicosis in Labrador Retrievers. Wu et al. investigated genetic modifiers of this disease [60]. Copper toxicosis is characterized by the accumulation of copper in the liver, eventually leading to cirrhosis. The authors performed NGS analysis on a cohort of 95 Labrador retrievers, analyzing 72 potential modifier genes for variations linked to hepatic copper levels [60]. Still not much is known regarding this disease, but the results so far suggest that conducting more such studies using NGS will help unravel this conundrum. Other authors performed whole-genome sequencing of a canine family trio revealing a *FAM83G* variant associated with hereditary footpad hyperkeratosis [47]. They found that the missense variant *FAM83G* is associated with this disease in Kromfohrländer dogs [47]. In addition, Yokoyama et al. found by using NGS a variation in genes related to cochlear biology which is strongly associated with adult-onset deafness in border collies [46]. All those studies suggest that using NGS in veterinary genetics will shed light on hereditary diseases. In addition, animals can be models for human genetic disease research. Lynos et al. performed whole-genome sequencing in cats, which allows to identify new models for blindness in AIPL1 and somite segmentation in HES7 [50]. 

In the future, NGS holds great promise for applications beyond its current uses in research and medical diagnostics, extending into fields such as forensic genetics and animal breeding. Particularly in the realm of canine and feline genetics, NGS could revolutionize the assessment of paternity and inheritance. By analyzing the complete genetic profiles of individuals, NGS will enable more accurate and detailed comparisons between offspring and potential parents, thereby enhancing the precision and reliability of paternity testing [25]. Furthermore, NGS could offer insights into complex inheritance patterns, including polygenic traits and genetic predispositions to diseases, facilitating informed breeding decisions aimed at improving the health, behavior, and characteristics of future generations of companion animals. As sequencing technologies continue to advance and become more accessible, the integration of NGS into routine genetic screening for canines and felines holds the potential to greatly benefit both breeders and pet owners alike.

## 5. Veterinary Parasitology

NGS has revolutionized many fields of biology, and veterinary parasitology is no exception. By allowing researchers to sequence DNA at an unprecedented speed and depth, NGS has opened up new avenues for understanding the diversity, evolution, and interactions of parasites that afflict animals.

Ballesteros et al. have investigated the first report in Columbia of *Dirofilaria repens* confirmed by NGS [61]. Interestingly, they undertook a study involving the PCR analysis of 100 serum samples, including the 12 previously identified as positive via the Giemsa-stained smear method. Surprisingly, none of these samples tested by PCR was positive for the specific dirofilarial species targeted. This phenomenon suggested the potential presence of nematode infections caused by other members of the *Onchocercidae* family, a crucial area for further investigation given its clinical relevance. Ultimately, the findings were corroborated through sequencing. This underscores the capability of NGS to address knowledge gaps that traditional PCR methods may not capture.

NGS can also find applications in classical parasitology. While routine laboratory microscopy examination offers certain advantages, literature reports indicate its limited effectiveness due to low-to-moderate diagnostic sensitivity [62]. Several authors have highlighted that the unsatisfactory diagnostic value of this exam stems from two common errors: processing and interpretation. Processing errors may occur at various stages, from field sampling to fecal material preparation in the laboratory, while misinterpretation can result from insufficient training of microscopy professionals in identifying parasite species broadly [63]. PCR is also available, but diagnostics for a single pathogen is possible when performing one reaction [64]. Serological methods are also available, but there is risk of false-positive results, because the antibodies could be increased a long time after infection or after just contact with the pathogen [65]. Due to those facts, NGS technology seems to be promising in the parasitology field. Lesniak et al. applied NGS in classical parasitology, i.e., when testing feces for parasites [66]. Their test allowed not only to demonstrate the presence of *Sarcocistis* spp. but also to determine their exact species in the tested samples. The determination of parasite species allowed the thesis of the spread of parasites between game and hunting dogs. NGS technologies could also be used to evaluate if the coinfection of few species of one pathogen occurs. Castillo-Castañeda et al. have evaluated many coinfections of *Trypanosomatidae* in samples in which infection has been previously diagnosed by PCR [67]. Kattor et al. used Targeted Next-Generation Sequencing for comprehensive testing for selected vector-borne pathogens in canines [68]. Their results showed false-negative results in two samples [68]. This method might be refined through further research.

## 6. Infectious Diseases in Animals

Infectious diseases pose significant threats to animal health, agricultural productivity, and even human health in cases of zoonotic transmission [21]. Traditional methods of diagnosing and studying these diseases often rely on culturing pathogens [69] or detecting specific antigens [70] or antibodies [71]. However, these approaches can be time-consuming, limited in sensitivity, and may fail to capture the full complexity of microbial communities involved in disease processes. Implement NGS, a transformative technology that offers unparalleled insights into the microbial world [72]. NGS technique enables the sequencing of microorganisms without requiring prior culture [24]. In addition, genetic characterization of infectious agents holds a pivotal position in the diagnosis, surveillance, and management of infectious diseases [73]. Li et al. used NGS technology to assist in the diagnosis of cat-scratch disease. They detected Bartonella henselae in cats with atypical symptoms and the diagnosis allows to recommend targeted therapy [74]. NGS offers powerful tools for exploring the intricate microbiomes present in diverse host species. Analysis of microbiome composition yields valuable insights into various facets of infection biology, including co-infection with multiple pathogens, the coexistence of viral and bacterial subpopulations within a single organ, and the evolutionary trajectories and genomic stability of pathogens. Such insights are instrumental in informing the development of innovative diagnostic methodologies and enhancing control strategies [75]. 

Charles et al. used this diagnostic tool to perform full diagnostic of etiology agents in dogs with tick-borne illness [76]. Blood and tick samples were collected from each dog and sent to a diagnostic laboratory for analysis. Microscopic examination revealed a mixed infection of intracytoplasmic organisms consistent with *Babesia* spp. and *Ehrlichia* spp. DNA extracted from the eight blood samples and 20 ticks underwent conventional PCR and NGS targeting the 16S rRNA and 18S rRNA genes for *Anaplasma/Ehrlichia* and *Babesia/Theileria/Hepatozoon*, respectively. *Ehrlichia* spp. DNA, closely resembling *Ehrlichia canis*, was identified in the blood of three dogs, while *Anaplasma* spp., closely related to *Anaplasma marginale*, was found in two dogs. *Babesia vogeli* was detected in one dog and seven ticks, while *Hepatozoon canis* and *Anaplasma* spp. were found in one tick [76]. Through NGS, the exact species of these pathogens and their phylogenetic origins could be determined. This is not the only research using NGS on tick-borne pathogens. Intirach et al. also obtained interesting results [77]. They performed tick’s pathogens screening based on PCR and NGS. Illumina NovaSeq sequencing showed a unique dominance of Bacteroidota in *Rhipicephalus* spp. [77]. This approach shed light into tick-borne illnesses. Davitt et al. also underscore the importance of pathogen surveillance studies in under-researched regions and reinforce the efficacy of NGS as an explorative diagnostic tool [78]. 

NGS is not only used in the diagnosis of tick-borne diseases. It is also finding practical applications in virology. The genome sequencing from canine clinical samples has been performed for Canine parvovirus (CPV) [79], Canine Distemper Virus (CDV) [80], and Canine morbillivirus [81]. Genomic sequencing for canine viruses holds paramount importance due to several key reasons. Firstly, it allows for the comprehensive characterization of viral strains, including their genetic diversity and evolution over time, which is crucial for understanding the epidemiology and transmission dynamics of these viruses. Additionally, genomic sequencing facilitates the identification of novel viral variants and mutations that may affect virulence, transmissibility, or vaccine efficacy, enabling timely adjustments to control and prevention strategies. Furthermore, it provides valuable insights into the genetic determinants of antiviral drug resistance, aiding in the development of effective treatment protocols. Overall, genomic sequencing plays a pivotal role in enhancing our ability to monitor, diagnose, and control canine viral diseases, ultimately safeguarding both animal and human health. Nowadays, despite the availability of a wide array of diagnostic tools, many conventional tests exhibit a high degree of specificity for particular viruses or target only a limited range of infectious agents [23]. This specificity presents obstacles or even barriers to the detection of new or unexpected pathogens, including emerging viruses or novel viral variants. However, the advent of viral metagenomics offers a promising and innovative approach to sample screening and virus detection, circumventing the need for prior knowledge of the infectious agent. NGS technology can be used in the detection of previously not-known infectious agents in various animal species [23]. An interesting finding was the detection of the Hepadnavirus in domestic cat (by using NGS in Japan [82]. In addition, the sequencing shows that that there are different subgroups of DCH, which opens the floodgates to a better understanding of this rare virus [82]. Momoi et al. used unbiased NGS to detect unknown viruses in cats [83]. Advanced nucleotide sequencing techniques have facilitated the detection and characterization of diverse viral genomes, encompassing pathogens such as the severe fever with thrombocytopenia syndrome virus, feline immunodeficiency virus, feline morbillivirus, parvovirus, and Torque teno felis virus [83].

NGS can also be used for clinical diagnostics. Boros et al. have found Unusual “Asian-origin” 2c to 2b point mutant canine Parvovirus and canine Astrovirus co-infection detected in vaccinated dogs with an outbreak of severe hemorrhagic gastroenteritis with high mortality rate by using NGS technologies [84]. Several studies using NGS on SARS-CoV-2 in dogs and cats have also been conducted. The infection confirmed by NGS technology has been evaluated and described in Botswana [85], Switzerland [86], the United States [87], and Spain [88]. Interestingly, the Spanish case presented hemorrhagic diarrhea with negative tests for candidate microbial pathogens [88]. Those studies affirm the need for monitoring dogs for possible clinically relevant SARS-CoV-2 transmissions between pets and humans [85]. It also shows that NGS could be used successfully in diagnostic non-obvious diseases in animals. 

## 7. Veterinary Immunology

Autoimmune diseases in animals, characterized by the immune system attacking the body’s own tissues [89], present diagnostic challenges similar to those in human medicine. These diseases can manifest in various forms and affect multiple organ systems, making accurate diagnosis essential for timely intervention and effective management [90]. While traditional diagnostic methods such as serological assays [91] and histopathology [92] provide valuable information, they may lack sensitivity and specificity, leading to delays in diagnosis or misclassification of diseases. Enter NGS, a revolutionary technology that offers comprehensive insights into the genetic underpinnings of autoimmune disorders.

In the field of immune diseases, not much research has been produced so far. Hwang et al. characterized canine immunoglobulin heavy-chain repertoire by NGS [93]. This study aimed to employ NGS for a thorough and unbiased analysis of the IGH repertoire in healthy dogs. Initially, the IGH locus was computationally searched for previously unidentified genes. Subsequently, IGH transcripts from major lymphoid organs were amplified using a 5′RACE approach, avoiding V/J primer bias. The resulting amplicons were sequenced on an Illumina MiSeq platform, and the data were analyzed using the ARResT/Interrogate platform. Analysis encompassed V/J usage, V-J pairing biases, isotype frequency, CDR3 diversity, convergent recombination, and public repertoires. The findings offer a comprehensive analysis of the IGH repertoire in healthy dogs, providing valuable insights for refining V/J gene-specific primer sets and establishing a baseline for future investigations into immune repertoires in both health and disease contexts [93].

## 8. Veterinary Oncology 

Cancer remains a significant threat to both human and animal health, posing complex challenges in diagnosis, treatment, and management [94]. Veterinary oncology, the study and treatment of cancer in animals, shares many similarities with its human counterpart, including the need for advanced molecular techniques to understand the underlying mechanisms of tumorigenesis and to develop targeted therapies [95]. NGS has emerged as a transformative technology in this field, offering unprecedented insights into the genomic landscape of tumors and paving the way for precision medicine approaches in veterinary oncology.

To advance understanding of canine cancers and facilitate the integration of personalized genomic analysis into clinical practice for canine cancer patients, Sakthikumar et al. have introduced SearchLight DNATM, a NGS gene panel [96]. This panel was designed to detect single-nucleotide variants, small insertions, or deletions this shortening has mentioned before, internal tandem duplications, and copy number variants across 120 cancer genes. The selection of these genes was meticulously curated based on their significant roles in driving the development and progression of both human and canine cancers, drawing from extensive review of primary canine literature and insights from human cancer databases. Validation of the panel encompassed diverse sample types, including formalin-fixed paraffin-embedded tissue, fine-needle aspirates, and fresh/frozen tissue [96]. Furthermore, the incorporation of structured biomarker annotation enhances the clinical utility of the panel, offering potential benefits in the diagnosis, prognostication, and treatment guidance for canine cancer patients. Through this endeavor, they have effectively pinpointed candidate pathogenic mutations, deepened our understanding of these diseases, and facilitated the necessary stratification for the advancement of novel therapeutic approaches [96].

However, NGS can find applications not only in research studies. It can also have applications in everyday veterinary practice. This technique may be useful in cancer early detection as a “liquid biopsy” test [97,98]. The advantages of this technique include simultaneously screening for multiple types of cancer with a simple blood test [97]. Apart from its role as a screening tool, liquid biopsy holds promise in canine patients as a diagnostic aid for cancer, particularly in scenarios where traditional diagnostic procedures pose challenges. In certain instances, cancer might be suspected, but pursuing a diagnosis through conventional methods, such as tissue sampling via surgical biopsy, could be impractical or deemed too hazardous due to factors like the mass’s anatomical location. Alternatively, cancer might rank high among the potential diagnoses based on clinical presentation, yet traditional diagnostic techniques are infeasible because a specific anatomical site for the suspected cancer is not clinically apparent [97].

## 9. Treatment Planning

In clinical practice, NGS can also be used when selecting an appropriate treatment method. Currently, antibiotic resistance of bacteria is high [99]. Carbapenemase-producing Enterobacterales (CPEs) pose a significant threat in human clinical environments, with instances of companion animal-origin CPEs sporadically reported in various countries. In their study, Harada et al. documented the first case of a canine infected with CPEs in Japan [100]. The patient, hospitalized due to pyometra, had pus from the uterus analyzed, revealing *Escherichia coli* resistant to several antibiotics but susceptible to others. Further testing indicated the presence of metallo-β-lactamases through the sodium mercaptoacetic acid double-disk synergy test. NGS identified the blaNDM-5 gene within an IncFII-type plasmid, alongside other resistance genes. Treatment with doxycycline and orbifloxacin proved successful in resolving the infection [100]. This case shows that in the absence of clinical improvement of the patient during antibiotic therapy, it is worth performing NGS to find the genes responsible for resistance to the drugs in question. Such a strategy will allow the selection of effective treatments. 

The advent of targeted NGS has revolutionized cancer diagnostics by offering sensitive genomic variant detection at affordable costs, ushering in a new era of personalized human oncology [101]. Given these advancements, it is plausible to anticipate the integration of targeted NGS into routine veterinary care in the near future. Numerous animal diseases share parallels with extensively studied human counterparts, suggesting that insights gleaned from human research could potentially illuminate the understanding and management of analogous conditions in animals [102]. We believe that, in the future, technologies based on NGS will allow using personalized treatment in veterinary oncology. 

## 10. Advantages and Disadvantages of NGS

The advantages of NGS are manifold, including its ability to generate vast amounts of data rapidly, allowing for comprehensive exploration of genomes, transcriptomes, and epigenomes (mapping of epigenetic modifications, such as DNA methylation, histone modifications, and chromatin accessibility, across the genome) [103]. NGS also facilitates the detection of genetic variants, including single-nucleotide polymorphisms (SNPs), insertions, deletions, and structural variations, with unparalleled sensitivity and resolution. Furthermore, NGS platforms offer flexibility in experimental design, accommodating diverse applications ranging from whole-genome sequencing to targeted resequencing and transcriptome profiling [7]. However, alongside these benefits, NGS comes with several limitations and challenges. These include issues related to data analysis and interpretation because of the fact that NGS generates massive amounts of data, necessitating sophisticated bioinformatics tools and computational resources for accurate processing and interpretation. Analyzing these data requires specialized expertise, and even with advanced algorithms, distinguishing true genetic variants from sequencing errors and technical artifacts can be challenging. Misinterpretation of data can lead to erroneous conclusions and hinder scientific progress [73]. NGS platforms can also introduce technical artifacts and biases during library preparation, sequencing, if sequencing depth is too low and during data processing steps. These artifacts may arise from PCR amplification biases, sequencing errors, or issues with base calling and alignment algorithms. Such biases can distort the representation of genetic variants in the sequencing data, complicating downstream analysis and potentially leading to false-positive or false-negative results [7,104]. While the cost of NGS has decreased significantly since its inception, the initial investment in instrumentation and ongoing expenses for reagents remain substantial. High-throughput sequencing instruments and consumables incur significant upfront costs, making them inaccessible to many researchers and clinicians, particularly those in resource-limited settings. Additionally, the need for regular maintenance and upgrades further contributes to the overall cost of implementing NGS technology [7,104]. Validating NGS results and establishing standardized protocols are critical for ensuring the reliability and reproducibility of findings. However, achieving consensus on best practices for sample preparation, sequencing, and data analysis can be daunting due to the diversity of NGS platforms, protocols, and analysis pipelines available. This lack of standardization complicates cross-study comparisons and hampers the integration of NGS data into clinical decision-making processes [68,73,97]. 

The integration of NGS technologies in animal research brings forth a myriad of ethical considerations that necessitate careful contemplation. While NGS offers unparalleled insights into the genetic makeup and biological processes of animals, ethical concerns arise regarding the welfare and treatment of research subjects. Fundamental principles of animal welfare, including the principles of replacement, reduction, and refinement (the 3Rs), must be upheld to minimize harm and ensure the ethical use of animals in research [105]. Moreover, issues related to informed consent, particularly in studies involving genetically modified animals or those bred for specific traits, warrant careful attention [106]. Additionally, the potential for unintended consequences, such as the generation of large datasets with implications beyond the scope of the original study, underscores the importance of responsible data management and dissemination [107]. Ethical oversight and regulatory frameworks must be robustly enforced to safeguard animal welfare and ensure the ethical conduct of NGS-driven research in animals. However, NGS probably remains the future for veterinary diagnostics. 

## 11. Future Perspectives for Sequencing 

The use of NGS has opened the floodgates for many new scientific studies and gained applications in medicine. However, with the development of sequencing technology, third-generation sequencing technology was developed. Third-generation sequencing technologies represent a paradigm shift in the field of genomics, offering unprecedented capabilities in sequencing long DNA fragments and overcoming many limitations of previous generations. These innovative platforms, such as single-molecule real-time (SMRT) sequencing and nanopore sequencing, enable direct observation of DNA synthesis in real time, eliminating the need for amplification and fragmentation steps [108,109]. This advancement not only facilitates the sequencing of complex genomic regions, including repetitive and structurally variable regions, but also allows for the detection of base modifications and DNA methylation patterns at single-molecule resolution. Moreover, third-generation sequencing platforms offer rapid turnaround times and reduced costs compared to traditional methods, making them increasingly accessible for a wide range of applications, including de novo genome assembly, metagenomics, and epigenetics research. As these technologies continue to evolve and improve, they hold great promise for revolutionizing our understanding of genomics and advancing precision medicine initiatives [13,110,111].

The MinIon and Sequel platforms represent two prominent third-generation sequencing technologies with distinct characteristics and performance profiles when compared to traditional NGS platforms [112,113]. In various scenarios, their relative performance can be evaluated based on factors such as read length, throughput, error rates, cost, and applicability, which are shown in Table 2.

## 12. Conclusions

In conclusion, the review of NGS techniques in veterinary medicine for companion animals underscores their transformative potential in diagnosis, treatment, and disease management. NGS offers unparalleled insights into the genetic makeup of animals, enabling precise identification of pathogens, characterization of genetic disorders, and personalized treatment strategies. As the field continues to advance, incorporating NGS into routine veterinary practice holds promise for improving animal health outcomes, enhancing breeding programs, and deepening our understanding of genetic diseases. However, challenges such as cost, data interpretation, and ethical considerations remain significant barriers to widespread adoption. Nevertheless, with ongoing research and technological advancements, NGS stands poised to revolutionize veterinary medicine, ushering in an era of precision healthcare for companion animals.

## Figures and Tables

**Figure 1 animals-14-01578-f001:**
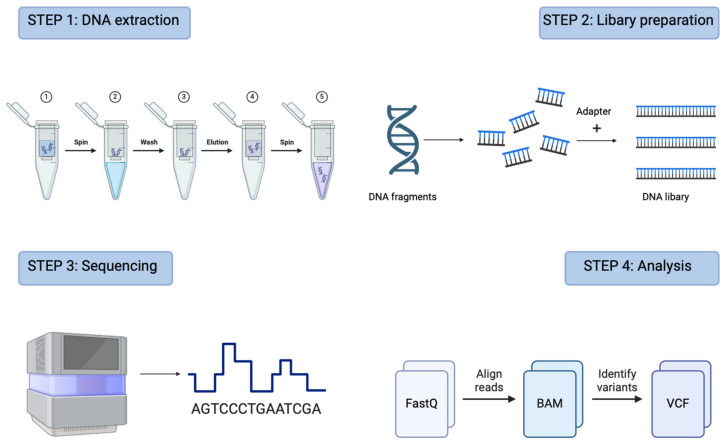
Four steps of NGS procedure. Figure generated by BioRender.

**Table 1 animals-14-01578-t001:** Types of sequencing and their advantages and disadvantages.

Type of Sequencing	Advantages	Disadvantages
Illumina Sequencing	High throughput, short read lengths, and low error rates make it ideal for applications requiring deep coverage and precise variant detection. It is cost-effective and widely accessible, with a broad range of library preparation kits available for various applications [4].	Limited read lengths, typically up to a few hundred base pairs, may pose challenges for de novo assembly and analysis of repetitive regions. PCR amplification during library preparation can introduce bias and errors, particularly in low-input samples [4].
Ion Torrent Sequencing	Offers rapid turnaround times and simple workflow, with no need for fluorescence labeling or imaging. It is suitable for targeted sequencing, amplicon sequencing, and small-genome sequencing [11].	Error rates are higher compared to other NGS platforms, particularly in homopolymeric regions. The detection of insertions and deletions (indels) can be challenging, leading to decreased accuracy in variant calling [11].
Pacific Biosciences (PacBio) Sequencing	Provides long read lengths, spanning thousands to tens of thousands of base pairs, facilitating de novo assembly, resolving complex genomic regions, and detecting structural variations with high precision. It also enables direct observation of epigenetic modifications, such as DNA methylation [12].	Higher error rates, particularly in the form of indels, limit the accuracy. Lower throughput and higher per-base costs may be prohibitive for large-scale projects requiring deep sequencing coverage [12]
Oxford Nanopore Sequencing:	Offers real-time sequencing, long read lengths, and portability, making it suitable for field applications, rapid diagnostics, and real-time monitoring of DNA synthesis. It does not require PCR amplification, allowing for direct sequencing of native DNA and RNA molecules [13].	Error rates can be relatively high, particularly in homopolymeric regions, although continuous improvements in base-calling algorithms are addressing this limitation. Additionally, the current cost per base is higher compared to other NGS platforms [13].
Long-Range Sequencing	Enables the detection of structural variations, haplotype phasing, and genome scaffolding by spanning large genomic distances. They complement short-read sequencing technologies and facilitate the assembly of complex genomes [14].	Limited to structural variant detection and genome scaffolding, long-range sequencing methods typically have lower throughput and higher costs compared to short-read sequencing platforms. Additionally, they may require additional library preparation steps and specialized instrumentation [14].

**Table 2 animals-14-01578-t002:** Comparison of two platforms MinIon and Sequel for third-generation sequencing with Sanger (NGS).

Platform	MinIon	Sequel	Sanger
Long-read sequencing	Offers long-read sequencing capabilities with read lengths ranging from thousands to tens of thousands of bases [112,114,115].	Provides longer average read lengths, often exceeding 10,000 bases [115,116].	Typically produces relatively short read lengths, ranging from several hundred to a few thousand bases [117].
Throughput	It has lower throughput than Sequel, processing a fewer number of reads per run. Its rapid sequencing turnaround time and low upfront cost can compensate for this limitation in certain applications [112].	Offers the highest throughput, generating a larger number of reads per run compared [112].	Relatively low-throughput and time-consuming, especially for projects requiring large-scale sequencing [6].
Error rates	Higher than Sequel [118]	Lower than MinIon [119]	Is considered the gold standard for accuracy, it has lower error rates compared to third-generation sequencing platforms [120].
Cost-effectiveness	Offers a lower than Sequel upfront cost and reduced instrument footprint. However, the cost per base can be relatively high due to consumable expenses [119].	Requires a higher than MinIon initial investment but may offer better cost-effectiveness in projects requiring large-scale sequencing due to its higher throughput and lower cost per base [121].	It is more cost-effective for targeted sequencing of individual genes or small genomic regions. However, for projects requiring whole-genome or whole-exome sequencing, the cost per base can become prohibitive [121].
Applicability:	Well-suited for rapid, real-time sequencing applications, field studies, and point-of-care diagnostics due to its portability and quick turnaround time [113,122].	Particularly advantageous for comprehensive genome analysis, including de novo assembly, structural variant detection, and long-range haplotyping, owing to its high throughput and long-read capabilities [122].	Widely used in clinical diagnostics for confirming genetic variants identified through other sequencing methods. It plays a crucial role in verifying pathogenic mutations, identifying rare variants, and assessing genetic heterogeneity in patients with inherited disorders or cancer [102].

## Data Availability

Not applicable.

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
