# Peer review of "Application of Next-Generation Sequencing (NGS) Techniques for Selected Companion Animals"

_animals, 2024, doi:10.3390/ani14111578_

Round 1

Reviewer 1 Report

Comments and Suggestions for Authors

The topic of the manuscript is targeting not only the field of veterinary medicine but also other areas. Unfortunately, there are many similar reviews and thus I would recommend the authors the following:

1) Do not limit your review to NGS but include Third generation sequencing as well.

2) Compare the relative performance of MinIon and Sequel with NGS platform for different scenarios (Lamb, Harrison J., et al. "The future of livestock management: a review of real-time portable sequencing applied to livestock." Genes 11.12 (2020): 1478.)

3) Extend the comparison also to non-massively parallel sequencing methods 

4) You did not cover the methylation at all (e.g. Schall, Peter Z., et al. "Genome-wide methylation patterns from canine nanopore assemblies." G3: Genes, Genomes, Genetics 13.11 (2023): jkad203., Perello, M. Godia, et al. "Nanopore long read sequencing for genome-wide cattle sperm methylation profiling." ISAG 2023: 39th International Society for Animal Genetics Conference. 2023, Kehl, Alexandra, et al. "Review of Molecular Technologies for Investigating Canine Cancer." Animals 14.5 (2024): 769, ...)

Author Response

Dear Reviewer 1, 

thank you for your time and valuable comments. All modifications in the main text have been added in yellow color.

  • Do not limit your review to NGS but include Third generation sequencing as well.

The data about third generation sequencing has been added into paragraph about future of sequencing.  Third-generation sequencing technologies represent a paradigm shift in the field of genomics, offering unprecedented capabilities in sequencing long DNA fragments and overcoming many limitations of previous generations. These innovative platforms, such as single-molecule real-time (SMRT) sequencing and nanopore sequencing, enable direct observation of DNA synthesis in real-time, eliminating the need for amplification and fragmentation steps [11,12]. This advancement not only facilitates the sequencing of complex genomic regions, including repetitive and structurally variable regions, but also allows for the detection of base modifications and DNA methylation patterns at single-molecule resolution. Moreover, third-generation sequencing platforms offer rapid turnaround times and reduced costs compared to traditional methods, making them increasingly accessible for a wide range of applications, including de novo genome assembly, metagenomics, and epigenetics research. As these technologies continue to evolve and improve, they hold great promise for revolutionizing our understanding of genomics and advancing precision medicine initiatives[13–15].

Third generation sequencing is not common in veterinary research and due to this fact we are unable to enrich this review with those data. In addition, this article focuses on NGS, not other sequencing methods. This interesting topic is also noteworthy. We are planning to focus on this topic in the future.

  • Compare the relative performance of MinIon and Sequel with NGS platform for different scenarios (Lamb, Harrison J., et al. "The future of livestock management: a review of real-time portable sequencing applied to livestock." Genes12 (2020): 1478.)

The comparison has been added into the table in paragraph about future of sequencing.  

  • Extend the comparison also to non-massively parallel sequencing methods 

It has been also included in the table 2

  • You did not cover the methylation at all (e.g.Schall, Peter Z., et al. "Genome-wide methylation patterns from canine nanopore assemblies." G3: Genes, Genomes, Genetics 11 (2023): jkad203., Perello, M. Godia, et al. "Nanopore long read sequencing for genome-wide cattle sperm methylation profiling." ISAG 2023: 39th International Society for Animal Genetics Conference. 2023, Kehl, Alexandra, et al. "Review of Molecular Technologies for Investigating Canine Cancer." Animals 14.5 (2024): 769, ...)

We didn’t mention about it because this article concern mainly on NGS, not on long read sequencing. If the reviewer wishes to enrich the article with this data, of course we can also do this.

Reviewer 2 Report

Comments and Suggestions for Authors

Dear Authors,

the topic of the manuscript is certainly of interest.

However, the application of a technology in such broad different, and complex fields and the discussion of its usefulness (or not) are inextricably based on knowledge and understanding of the technologies itself, as well as the fields in which they are applied. This is necessary not to miss the point.

Since its inception, this work has presented incorrectness and inaccuracies. Phrasing and references are quite imprecise and sometimes inappropriate, mainly in the introduction section. 

The state of art fails to correctly contextualize and clarify what is next-generation sequencing. Throughout the manuscript, there is some apparent confusion between the concepts of high-throughput, genome-wide, and next-generation sequencing.

The introduction must be revised/rewritten about the molecular biology or genetic aspects.

The title refers to Selected Companion Animals, but all is almost about dogs, only despite next-generation sequencing being widely applied to cats too. In the manuscript, out of 92 references REV counted only two references (n.4 and 81) about specific studies on cats (both about microbiota), in Ref 52, the work is about RNA seq in dogs and several other species (including the cat) data are considered from other papers for comparison. In Ref 55 cat together with other species is part of a multispecies STR panel for forensics. In lines 189, 248, and 257 cat is just mentioned.

The reviewer's suggestion is changing the title to “selected dogs” or increasing the examples/references specifically to cats.

Throughout the manuscript no clear distinction is made between research, where NGS is often an integration (as correctly mentioned, but not made explicit), and routine or diagnostic applications (still few, but existing). For instance, in lines 115-117, this distinction is made but practical application is cited without a reference. This distinction could be made in the introduction.

 Overall the choice of references seems sometimes random.

Some suggestions:

AA Line 39-42: “In recent years, a significant amount of research has been based on the polymerase chain reaction (PCR) and its variations [1]”. 

REV [Ref.1* = ristensen LS, Hansen LL. PCR-based methods for detecting single-locus DNA methylation biomarkers in cancer diagnostics, prognostics, and response to treatment. Clin Chem. 2009 Aug;55(8):1471-83. doi: 10.1373/clinchem.2008.121962. Epub 2009 Jun 11. PMID: 19520761.] PCR dates back to 1983 by Mullis, not recent years. The authors cite a quite recent review on a specific application to introduce PCR that is the foundation of nucleic acid studies of the last 40 years in every biological field. Please change the reference 

AA Lines 40-42 PCR is still used a lot in everyday veterinary practice because clinicians have access to the technology and the test is relatively inexpensive”

REV. PCR is a molecular method that allows the specific amplification of a small DNA region. Which “test” are the authors talking about? Not precise terminology, please rephrase.

“Still used”: so what do the authors mean by PCR in this introduction? An amplification very close to the PCR principle itself (two primers, cycled amplification, SS template, and a polymerase ) is used also in the Next (second) generation sequencing. Here PCR term is used in jargony and approximate way. Please rephrase.

AA Lines 46-47: “One primary advantage of this method is its capability for real-time sequencing”

REV in molecular biology real-time sequencing refers to a so-called third-generation sequencing method developed in the last 10 years mainly by PacBio. It is a 1) long reads 2) real-time 3) quantitative method. It is very different from classic first (Sanger) and second-generation sequencing which both are not “real-time”, not long reads and not quantitative methods. So what does the AA mean with real-time sequencing? Please rephrase

Lines 48-49 “This approach boasts rapidity, being approximately 100 times faster than the chain termination method, and is fully automated [3]

REV Not clear what the authors are talking about. Perhaps they refer to the automated Sanger sequencing with fluorescent ddNTPs (modified Sanger), released in the middle of the ‘90s. It is faster because more samples can be co-run into one single gel, not because is not based on chain termination. It is also safer because uses fluorochromes instead of radioactivity. Ref [3] is about pyrosequencing (following sentence), not Sanger".

Lines 49-51Pyrosequencing, emerging as a result of these advancements, laid the groundwork for a novel sequencing paradigm later dubbed Next Generation Sequencing (NGS) [4].”

REV. This is incomplete. Massive parallel sequencing or Next-generation sequencing (NGS) or second-generation sequencing is characterized by two related terms: - massively parallel (simultaneously in the same manner) and deep sequencing. To do that different NGS platforms are using different sequencing technologies. Pyrosequencing laid the groundwork for one of those. The REV had no access to the review Ref 4 to verify the AA sentence.

Line 56Diverging from traditional Sanger sequencing, which demands a uniform DNA template”

REV Perhaps does AA mean “unique template”?

Lines 56-58 “NGS facilitates the examination of heterogeneous samples while concurrently furnishing sequence data for over 10 million randomly chosen nucleic acid molecules within a sample [6].

REV please rephrase. It is not clear what the AA means by “randomly chosen”. Certainly, with the addition of index/barcodes in the amplification phase, it is possible to sequence different higher organisms at the same time, it is certainly possible to obtain laws that cover many times thousands of targeted regions also of heterogeneous templates, such as whole genome sequence with deep coverage and depth,  but I don't understand the context of “randomly choosing molecular acid molecules” and I haven't found the citation in the reference.

Lines 68 – 70 “The approach to library preparation and the selection of sequencing instruments and associated technologies significantly influences the potential downstream analyses [10].”

REV Ref10 is a review of NGS in the microbiological world. Is this valid in all the fields as here the manuscript is generally speaking and mainly about mammals?

Lines 89-92 From identifying genetic predispositions to diseases [28–30] and assessing microbial diversity in animal populations [26,31,32] to monitoring the spread of infectious agents and improving breeding programs, NGS offers unprecedented insights that are reshaping the landscape of veterinary diagnostics and research [33–35].

REV Overall, but particularly in “Predisposition to diseases”, in this section should be better to cite reviews instead of single papers, which can be eventually mentioned in the following specific paragraphs.

Lines 195-198

REV. “Advancements in veterinary genetics have transformed our understanding of hereditary traits, diseases, and population dynamics in animals [28,29,47]. The integration of 196 Next-Generation Sequencing technologies has been instrumental in accelerating progress 197 in this field.“. Ref 28 and 29 refer to hereditary diseases, so 1) include a reference to a trait, such as coat color or tail, or 2) exclude the term traits and only mention hereditary diseases. Both the cited manuscripts have a first high-throughput genome-wide mapping step (GWAS, genome-wide association study using microarrays, not sequencing) and then a targeted sequencing to refine. The authors correctly mention here the “integration” of NGS technology but the nature of this integration remains obscure to the reader because it has not been clarified the terminology, purpose, and methods as mentioned above.

Ref. 48 is a review about obesity. Population dynamics is defined as “the portion of ecology that deals with the variation in time and space of population size and density for one or more species (Begon et al. 1990).” “The four vital rates of population dynamics for measuring population growth are birth, death, immigration, and emigration” I do not think ref.48 fits the topic of next-gen in population dynamics.

Lines 226-246

REV. Authors cited nowhere the well-established canine and feline genome projects, nor OMIA.

Line 226 “Many diseases have a genetic basis and are inherited [57]. “ To support this sentence is cited a paper dated 2000 on a mailed questionnaire to survey the feelings of  Finnish Dog-Owners about Programs to Control Canine 645 Genetic Diseases” (n.57: Leppänen, M.; Paloheimo, A.; Saloniemi, H. Attitudes of Finnish Dog-Owners about Programs to Control Canine 645 Genetic Diseases. Prev Vet Med 2000, 43, 145–158, doi:10.1016/S0167-5877(99)00098-7.)

Lines 248-251 “Particularly in the realm of canine and feline genetics, NGS could revolutionize the assessment of paternity and inheritance. By analyzing the complete genetic profiles of individuals, NGS will enable more accurate and detailed comparisons between offspring and potential parents, thereby enhancing the precision and reliability of paternity testing.”

REV This sentence needs a reference.

Author Response

Dear Reviewer 2, 

thank you for your time and valuable comments. All modifications in the main text have been added in green color.

The introduction must be revised/rewritten about the molecular biology or genetic aspects. -the genetic aspects are discussed in paragraph Veterinary genetic

The title refers to Selected Companion Animals, but all is almost about dogs, only despite next-generation sequencing being widely applied to cats too. In the manuscript, out of 92 references REV counted only two references (n.4 and 81) about specific studies on cats (both about microbiota), in Ref 52, the work is about RNA seq in dogs and several other species (including the cat) data are considered from other papers for comparison. In Ref 55 cat together with other species is part of a multispecies STR panel for forensics. In lines 189, 248, and 257 cat is just mentioned. 

The reviewer's suggestion is changing the title to “selected dogs” or increasing the examples/references specifically to cats.

 The data about cats has been added:

  • In metabolic paragraph: Rodney et al., evaluated new gene candidates, notably detecting novel missense variants associated with polycystic kidney disease in the Peterbald cat breed. These findings underscore the efficacy of Whole Exome Sequencing (WES) in pinpointing previously unknown gene candidate alleles linked to diseases and traits, marking a significant advancement in feline model research [41].
  • In genetic: Few authors performed whole feline genome sequencing [46–49]. Sugasawa et al., established a sequencing method for the whole mitochondrial DNA of domestic dogs [26].

And: All those studies suggests that using NGS in veterinary genetic will shed a light on hereditary diseasess. In addition, animals can be models for human genetic disease research. Lynos et al., performed whole genome sequencing in cats, which allows to identify new models for blindness in AIPL1 and somite segmentation in HES7 [46].

  • In infectious diseases: In addition, genetic characterization of infectious agents holds a pivotal position in the diagnosis, surveillance, and management of infectious diseases [65]. Li et al., used NGS technology to assist diagnosis of cat-scratch disease. They detected Bartonella henselae in cat with atypical symptoms and the diagnosis allows to recommendation targeted therapy[66].

And: An interesting finding was detection of domestic cat hepadnavirus (by using NGS in Japan [74]. In addition, the sequencing shows that that there are different subgroups of DCH which opens the floodgates to a better understanding of this rare virus [74]. Momoi et al., used unbiased NGS to detect unknown viruses in cats [75]. Advanced nucleotide sequencing techniques have facilitated the detection and characterization of diverse viral genomes, encompassing pathogens such as the severe fever with thrombocytopenia syndrome virus, feline immunodeficiency virus, feline morbillivirus, parvovirus, and Torque teno felis virus[75].

And: Several studies using NGS on SARS-CoV-2 in dogs and cats have been also conducted. The infection confirmed by NGS technology has been evaluated and described in Botswana [77],  Switzerland [78]  United States[79] and Spain [80].

Throughout the manuscript no clear distinction is made between research, where NGS is often an integration (as correctly mentioned, but not made explicit), and routine or diagnostic applications (still few, but existing). For instance, in lines 115-117, this distinction is made but practical application is cited without a reference. This distinction could be made in the introduction. -the citation has been added

 Overall the choice of references seems sometimes random.

Some suggestions: 

AA Line 39-42: “In recent years, a significant amount of research has been based on the polymerase chain reaction (PCR) and its variations [1]”.  

REV [Ref.1* = ristensen LS, Hansen LL. PCR-based methods for detecting single-locus DNA methylation biomarkers in cancer diagnostics, prognostics, and response to treatment. Clin Chem. 2009 Aug;55(8):1471-83. doi: 10.1373/clinchem.2008.121962. Epub 2009 Jun 11. PMID: 19520761.] PCR dates back to 1983 by Mullis, not recent years. The authors cite a quite recent review on a specific application to introduce PCR that is the foundation of nucleic acid studies of the last 40 years in every biological field. Please change the reference  - it has been changed

AA Lines 40-42  PCR is still used a lot in everyday veterinary practice because clinicians have access to the technology and the test is relatively inexpensive” 
REV. PCR is a molecular method that allows the specific amplification of a small DNA region. Which “test” are the authors talking about? Not precise terminology, please rephrase.  it has been precised

“Still used”: so what do the authors mean by PCR in this introduction? An amplification very close to the PCR principle itself (two primers, cycled amplification, SS template, and a polymerase ) is used also in the Next (second) generation sequencing. Here PCR term is used in jargony and approximate way. Please rephrase. – it has been precised

AA Lines 46-47: “One primary advantage of this method is its capability for real-time sequencing” 
REV in molecular biology real-time sequencing refers to a so-called third-generation sequencing method developed in the last 10 years mainly by PacBio. It is a 1) long reads 2) real-time 3) quantitative method. It is very different from classic first (Sanger) and second-generation sequencing which both are not “real-time”, not long reads and not quantitative methods. So what does the AA mean with real-time sequencing? Please rephrase -it has been rephrased

Lines 48-49 “This approach boasts rapidity, being approximately 100 times faster than the chain termination method, and is fully automated [3]          
REV Not clear what the authors are talking about. Perhaps they refer to the automated Sanger sequencing with fluorescent ddNTPs (modified Sanger), released in the middle of the ‘90s. It is faster because more samples can be co-run into one single gel, not because is not based on chain termination. It is also safer because uses fluorochromes instead of radioactivity. Ref [3] is about pyrosequencing (following sentence), not Sanger".it has been clarified and the reference has been changed

Lines 49-51 Pyrosequencing, emerging as a result of these advancements, laid the groundwork for a novel sequencing paradigm later dubbed Next Generation Sequencing (NGS) [4].” 
REV. This is incomplete. Massive parallel sequencing or Next-generation sequencing (NGS) or second-generation sequencing is characterized by two related terms: - massively parallel (simultaneously in the same manner) and deep sequencing. To do that different NGS platforms are using different sequencing technologies. Pyrosequencing laid the groundwork for one of those. The REV had no access to the review Ref 4 to verify the AA sentence. – it has been verified

Line 56 Diverging from traditional Sanger sequencing, which demands a uniform DNA template” 
REV Perhaps does AA mean “unique template”? -yes, it has been corrected

Lines 56-58 “NGS facilitates the examination of heterogeneous samples while concurrently furnishing sequence data for over 10 million randomly chosen nucleic acid molecules within a sample [6]. 
REV please rephrase. It is not clear what the AA means by “randomly chosen”. Certainly, with the addition of index/barcodes in the amplification phase, it is possible to sequence different higher organisms at the same time, it is certainly possible to obtain laws that cover many times thousands of targeted regions also of heterogeneous templates, such as whole genome sequence with deep coverage and depth,  but I don't understand the context of “randomly choosing molecular acid molecules” and I haven't found the citation in the reference.- the term randomly chosen has been deleted, the citation has been corrected

Lines 68 – 70 “The approach to library preparation and the selection of sequencing instruments and associated technologies significantly influences the potential downstream analyses [10].” 
REV Ref10 is a review of NGS in the microbiological world. Is this valid in all the fields as here the manuscript is generally speaking and mainly about mammals? - the citation has been corrected

Lines 89-92 From identifying genetic predispositions to diseases [28–30] and assessing microbial diversity in animal populations [26,31,32] to monitoring the spread of infectious agents and improving breeding programs, NGS offers unprecedented insights that are reshaping the landscape of veterinary diagnostics and research [33–35].          
REV Overall, but particularly in “Predisposition to diseases”, in this section should be better to cite reviews instead of single papers, which can be eventually mentioned in the following specific paragraphs. - the citation has been corrected

Lines 195-198 
REV. “Advancements in veterinary genetics have transformed our understanding of hereditary traits, diseases, and population dynamics in animals [28,29,47]. The integration of 196 Next-Generation Sequencing technologies has been instrumental in accelerating progress 197 in this field.“.
Ref 28 and 29 refer to hereditary diseases, so 1) include a reference to a trait, such as coat color or tail, or 2) exclude the term traits and only mention hereditary diseases. Both the cited manuscripts have a first high-throughput genome-wide mapping step (GWAS, genome-wide association study using microarrays, not sequencing) and then a targeted sequencing to refine. The authors correctly mention here the “integration” of NGS technology but the nature of this integration remains obscure to the reader because it has not been clarified the terminology, purpose, and methods as mentioned above. term traits have been deleted

Ref. 48 is a review about obesity. Population dynamics is defined as “the portion of ecology that deals with the variation in time and space of population size and density for one or more species (Begon et al. 1990).” “The four vital rates of population dynamics for measuring population growth are birth, death, immigration, and emigration” I do not think ref.48 fits the topic of next-gen in population dynamics. -this citation has been deleted

Lines 226-246  
REV. Authors cited nowhere the well-established canine and feline genome projects, nor OMIA. 
The paragraph has been added: Several well-established genome projects have significantly advanced our understanding of canine and feline genetics, providing invaluable resources for both veterinary and comparative genomic research. The Canine Genome Project, stands as one of the pioneering efforts in this domain[50]. Led by an international consortium of researchers, this project aimed to sequence and annotate the entire canine genome, resulting in the publication of the first draft assembly in 2005[51]. Since then, continual improvements in sequencing technologies and bioinformatics tools have facilitated the refinement of the dog genome assembly, enabling comprehensive analyses of genetic variation, breed diversity, and disease susceptibility across canine populations. Similarly, the Feline Genome Project[52], launched with the goal of elucidating the genetic makeup of domestic cats, has made significant strides in characterizing the feline genome. Through collaborative efforts and advancements in sequencing methodologies, researchers have produced high-quality genome assemblies for various cat breeds, laying the foundation for investigating the genetic basis of feline traits, diseases, and evolutionary adaptations. These genome projects serve as invaluable resources for deciphering the genetic underpinnings of complex traits, facilitating the development of diagnostic tools, and informing breeding practices aimed at promoting animal health and welfare [50–52].

Line 226 “Many diseases have a genetic basis and are inherited [57]. “ To support this sentence is cited a paper dated 2000 on a mailed questionnaire to survey the feelings of  Finnish Dog-Owners about Programs to Control Canine 645 Genetic Diseases” (n.57: Leppänen, M.; Paloheimo, A.; Saloniemi, H. Attitudes of Finnish Dog-Owners about Programs to Control Canine 645 Genetic Diseases. Prev Vet Med 2000, 43, 145–158, doi:10.1016/S0167-5877(99)00098-7.) – new citation has been added

Lines 248-251 “Particularly in the realm of canine and feline genetics, NGS could revolutionize the assessment of paternity and inheritance. By analyzing the complete genetic profiles of individuals, NGS will enable more accurate and detailed comparisons between offspring and potential parents, thereby enhancing the precision and reliability of paternity testing.”         
REV This sentence needs a reference. -the citation has been added     

Reviewer 3 Report

Comments and Suggestions for Authors

The manuscript “Application of Next-Generation Sequencing (NGS) Techniques for

Selected Companion Animals“ provides a general overview on the applications of NGS in veterinary medicine and therefore is of interest for the advanced veterinarian.

Major revisions

General: The information in each section is interesting, but not always well presented. The text of the manuscript is too close to the abstract of the papers cited at some points. The review should also be better structured, to make the included information more easily digestible. I suggest to provide the question of the study, type of analysis (e.g. expression analysis, whole genome analysis, gene panel etc.), sampling material used, NGS method used and the essence of the outcome (genetic pathway X was upregulated / the gene Y, responsible for Z-disease showed a causal mutation etc.). The less relevant information should be thoroughly trimmed throughout the complete manuscript. Don’t get lost in the details.

Abstract: The animal species the review focuses on should be named.

Lines 46-47: It seems that in this passage methods get mixed up. Sanger sequencing is per definition a chain termination method, from which pyrosequencing differs. It would also be desirable to have a short overview on the different NGS sequencing methods, which are mainly currently used in the next passages (including long-range sequencing), with their individual advantages and disadvantages. In addition, as the standard veterinarian might not be used to the different types of NGS analyses (e.g. transcriptome, genome, epigenome…), there should be a short explanation of them with their potential applications.

Line 93 ff: In my view, topic 2 should better be separated into two different topics: “Reproduction research” or something alike (lines 94-118) and “Microbiology” (lines 95-143), because the underlying studies also differ in terms of their NGS methodology. In addition, the paragraph around lines 101-118 can be significantly shortened and should be reduced to its essence.

Topic 4: Veterinary genetics: It should be made very clear, that there is an overwhelming multitude of studies including whole-genome sequencing on various breeds. For each application (monogenic variant, polygenic/modifier genes, breed sequencing etc.) there should be more examples, but don't dwell too long on a single publication. Just one example: in line 236, its said that 44 variants were detected in 25 genes initially. This information is unnecessary, as it would be sufficient to mention that four variants showed significant association with copper level. Many of the descriptions can be shortened like this.

Minor revisions

General: Quotations should be continuous within the text, not mixed up.

If you cite with author names, it should be without comma following the name  

As the shortcut NGS has been explained at the beginning, it should be used throughout the text instead of “next-generation sequencing”

Lines 32 ff: Please phrase more carefully, as currently, this is not typically handled that way yet.

Line 39: I guess the term “recent” is not appropriate, as PCR techniques in general have been used for quite a long time.

Line 120: Change “had” to “have”

Line 122: Change “evaluation” to “evaluate”

Line 122: This sentence fits better to line 141, behind the studies on allergic vs. non-allergic dogs.

Line 159: Replace “found that suggests” by “suggested”

Lines 171-172: Delete sentence as this is not specific to metabolic processes. It would rather fit into the introduction.

Line 176: gene expression

Line 184: Replace “In study Uno” by “In a study by Uno”

Line 201: “A study performed”

Line 207-208: Delete sentence because its too general. Maybe there is a more specific perspective?

Line 229: Replace “diseased” by “affected”

Line 229: Replace “One of genetic” by “One example for genetic”

Line 239: Replace Trio Reveals” to “trio reveals” and “wenome” to “genome”

Line 296: Start this last sentence with “This method might be refined…”

Line 304: Start sentence with “This method might be refined…”

Line 317: “tick-borne”

Lines 333-334: Delete this sentence, as the connection with NGS is not clear

Line 364 ff: Rephrase as it's not entirely up-to-date anymore.

Line 368: I would suggest Immunology as header.

Line 465: Sequencing artifacts and errors are mainly an issue if sequencing depth is too low.

Line 493: These ethical considerations should also be added to the Section Number 10 (Advantages/disadvantages)

Comments on the Quality of English Language

I have no comments on quality of English language.

Author Response

Dear Reviewer 3, 

thank you for your time and valuable comments. All modifications in the main text have been added in red color.

General: The information in each section is interesting, but not always well presented. The text of the manuscript is too close to the abstract of the papers cited at some points.-it has been rewritten

 The review should also be better structured, to make the included information more easily digestible. I suggest to provide the question of the study, type of analysis (e.g. expression analysis, whole genome analysis, gene panel etc.), sampling material used, NGS method used and the essence of the outcome (genetic pathway X was upregulated / the gene Y, responsible for Z-disease showed a causal mutation etc.). The less relevant information should be thoroughly trimmed throughout the complete manuscript. Don’t get lost in the details. - Addressing the suggested revision would lead to the inclusion of descriptions of individual authors' experiences, which may not align with the overarching theme of this article. Our primary focus is on the applications of this methodology in science and clinical settings, which have been relatively limited thus far. Consequently, the scope of the research objectives and methodology outlined in the study is restricted, with emphasis placed on highlighting the accomplishments attained through this research approach.

Abstract: The animal species the review focuses on should be named.  it has been added

Lines 46-47: It seems that in this passage methods get mixed up. Sanger sequencing is per definition a chain termination method, from which pyrosequencing differs. It would also be desirable to have a short overview on the different NGS sequencing methods, which are mainly currently used in the next passages (including long-range sequencing), with their individual advantages and disadvantages. - a brief overview of the main NGS sequencing methods currently used, along with their individual advantages and disadvantages has been added in the table 1.

 In addition, as the standard veterinarian might not be used to the different types of NGS analyses (e.g. transcriptome, genome, epigenome…), there should be a short explanation of them with their potential applications. –

All those terms has been explained:

  • In their study, Nowak et al. sought to deepen understanding of the mechanisms underlying the termination of canine pregnancy by employing transciptome sequencig [30],which involves the high-throughput sequencing of RNA molecules to quantify gene expression levels and identify transcript isoforms[31]. They analyzed placental transcriptomes derived from natural prepartum and antigestagen-induced abortions and compared them with fully developed mid-gestation placentas.
  • Few authors performed whole feline genome sequencing [50–53] which involves the determination of the complete DNA sequence of an organism's genome, including all its genes and non-coding regions[54].
  • The advantages of NGS are manifold, including its ability to generate vast amounts of data rapidly, allowing for comprehensive exploration of genomes, transcriptomes, and epigenomes (mapping of epigenetic modifications, such as DNA methylation, histone modifications, and chromatin accessibility, across the genome) [105].

Line 93 ff: In my view, topic 2 should better be separated into two different topics: “Reproduction research” or something alike (lines 94-118) and “Microbiology” (lines 95-143), because the underlying studies also differ in terms of their NGS methodology. – The paragraph has been divided into two subheadings

 In addition, the paragraph around lines 101-118 can be significantly shortened and should be reduced to its essence.- It has been shortened

Topic 4: Veterinary genetics: It should be made very clear, that there is an overwhelming multitude of studies including whole-genome sequencing on various breeds. For each application (monogenic variant, polygenic/modifier genes, breed sequencing etc.) there should be more examples, but don't dwell too long on a single publication. Just one example: in line 236, its said that 44 variants were detected in 25 genes initially. This information is unnecessary, as it would be sufficient to mention that four variants showed significant association with copper level. Many of the descriptions can be shortened like this. – this paragraph has been shortened

Minor revisions

General: Quotations should be continuous within the text, not mixed up. If you cite with author names, it should be without comma following the name – comas have been deleted

As the shortcut NGS has been explained at the beginning, it should be used throughout the text instead of “next-generation sequencing” -It has been corrected

Lines 32 ff: Please phrase more carefully, as currently, this is not typically handled that way yet.

This sentence has been rewritten: “Application of NGS in treatment planning enhances precision and efficacy by enabling personalized therapeutic strategies tailored to individual animals and its diseases, ultimately advancing veterinary care for companion animals.”

Line 39: I guess the term “recent” is not appropriate, as PCR techniques in general have been used for quite a long time. It has been changed for: In the preceding years

Line 120: Change “had” to “have” -It has been corrected

Line 122: Change “evaluation” to “evaluate” -It has been corrected

Line 122: This sentence fits better to line 141, behind the studies on allergic vs. non-allergic dogs. -It has been corrected

Line 159: Replace “found that suggests” by “suggested” -It has been corrected

Lines 171-172: Delete sentence as this is not specific to metabolic processes. It would rather fit into the introduction. -It has been deleted

Line 176: gene expression -It has been corrected

Line 184: Replace “In study Uno” by “In a study by Uno” -It has been corrected

Line 201: “A study performed” -It has been corrected

Line 207-208: Delete sentence because its too general. Maybe there is a more specific perspective? – it has been deleted

Line 229: Replace “diseased” by “affected” -It has been corrected

Line 229: Replace “One of genetic” by “One example for genetic” -It has been corrected

Line 239: Replace Trio Reveals” to “trio reveals” and “wenome” to “genome” -It has been corrected

Line 296: Start this last sentence with “This method might be refined…” -It has been corrected

Line 317: “tick-borne” -It has been corrected

Lines 333-334: Delete this sentence, as the connection with NGS is not clearIt has been deleted

Line 364 ff: Rephrase as it's not entirely up-to-date anymore. – It has been rephrased: Those studies affirms the need for monitoring dogs for possible clinically relevant SARS-CoV-2 transmissions between pets and humans [77]

Line 368: I would suggest Immunology as header. -It has been corrected

Line 465: Sequencing artifacts and errors are mainly an issue if sequencing depth is too low. -This information has been added.

Line 493: These ethical considerations should also be added to the Section Number 10 (Advantages/disadvantages)-ethical considerations has been added:

The integration of NGS technologies in animal research brings forth a myriad of ethical considerations that necessitate careful contemplation. While NGS offers unparalleled insights into the genetic makeup and biological processes of animals, ethical concerns arise regarding the welfare and treatment of research subjects. Fundamental principles of animal welfare, including the principles of replacement, reduction, and refinement (the 3Rs), must be upheld to minimize harm and ensure the ethical use of animals in research[108]. Moreover, issues related to informed consent, particularly in studies involving genetically modified animals or those bred for specific traits, warrant careful attention [109]. Additionally, the potential for unintended consequences, such as the generation of large datasets with implications beyond the scope of the original study, underscores the importance of responsible data management and dissemination[110]. Ethical oversight and regulatory frameworks must be robustly enforced to safeguard animal welfare and ensure the ethical conduct of NGS-driven research in animals.

Round 2

Reviewer 2 Report

Comments and Suggestions for Authors

Dear Authors,

The manuscript is improved.

A few minor comments need to be addressed.

New lines 41

AA: “Real-time and RT-PCR”.

REV. Generally, Real-Time is shortly named RT-PCR or qPCR, synonyms. Please fix

New lines 48-50 - 1st vs Lines 48-49 - it has been clarified and the reference has been changed

AA “Fred Sanger, who developed a sequencing technique relying on polynucleotide chain termination [3]. One primary advantage of this method is its capability for sequencing, wherein the result is obtained concurrently with the synthesis of the new DNA strand. This approach boasts rapidity, being approximately 100 times faster than the chain termination method, because more samples can be co-run into one single gel, and is fully automated[3].

REV: What is “this approach” referred to?

If it is referred to the modified fluorescent Sanger method the sentence is still incorrect.

Classical Sanger using radioactive ddNTPs (four lanes/sample and manual loading each run) and automatized Sanger with fluorescent ddNTPs (one capillary/sample and automated loading of multiple subsequent runs)  both are chain terminator methods [https://en.wikipedia.org/wiki/Sanger_sequencing].

The reason why the automatized system is “faster” than the classic one is because of the automatized loading, of the fluorescent ddNTPs and also the connected system (PC, etc), but both are chain terminator methods.

Please I replied in detail to be as clear as possible, but these details are not all necessary in the manuscript However the sentence in lines 48-50 must be rephrased and reference n.3 is incorrect and needs to be changed: the first modified systems were released in the ‘90s.

New Line 252  (Old Line 226)

AA. Many diseases have a genetic basis and are inherited [57]. “ To support this sentence is cited a paper dated 2000 on a mailed questionnaire to survey the feelings of Finnish Dog-Owners about Programs to Control Canine 645 Genetic Diseases” (n.57: Leppänen, M.; Paloheimo, A.; Saloniemi, H. Attitudes of Finnish Dog-Owners about Programs to Control Canine 645 Genetic Diseases. Prev Vet Med 2000, 43, 145–158, doi:10.1016/S0167-5877(99)00098-7.) – A new citation has been added.

REV This added paper is more about genetic counseling (such as the previous n.57 ) and inbreeding control. They are neither centered nor meaningful to the topic. Here citation of OMIA (whose citation has already been suggested) includes everything. Please, exclude the two cited papers (or if you strongly need to leave them) but add: Nicholas, F.W., Tammen, I., & Sydney Informatics Hub. (1995). Online Mendelian Inheritance in Animals (OMIA) [dataset]. https://omia.org/https://doi.org/10.25910/2AMR-PV70. The citation date is 1995, but the site is constantly updated (https://omia.org/home/)

Author Response

Dear Reviewer 2, 

thank you for your time and valuable comments. All modifications in the main text have been added in green color.

New lines 41

AA: “Real-time and RT-PCR”. 

REV. Generally, Real-Time is shortly named RT-PCR or qPCR, synonyms. Please fix -it has been fixed:

Real-time PCR (RT-PCR) is still used a lot in everyday veterinary practice, because clinicians have access to the technology and the test is relatively inexpensive.

New lines 48-50 - 1st vs Lines 48-49 - it has been clarified and the reference has been changed

AA “Fred Sanger, who developed a sequencing technique relying on polynucleotide chain termination [3]. One primary advantage of this method is its capability for sequencing, wherein the result is obtained concurrently with the synthesis of the new DNA strand. This approach boasts rapidity, being approximately 100 times faster than the chain termination method, because more samples can be co-run into one single gel, and is fully automated[3].

REV: What is “this approach” referred to?

If it is referred to the modified fluorescent Sanger method the sentence is still incorrect. 

Classical Sanger using radioactive ddNTPs (four lanes/sample and manual loading each run) and automatized Sanger with fluorescent ddNTPs (one capillary/sample and automated loading of multiple subsequent runs)  both are chain terminator methods [https://en.wikipedia.org/wiki/Sanger_sequencing]. 

The reason why the automatized system is “faster” than the classic one is because of the automatized loading, of the fluorescent ddNTPs and also the connected system (PC, etc), but both are chain terminator methods. -it has been rephrased:

One primary advantage of this method is its capability for sequencing, wherein the result is obtained concurrently with the synthesis of the new DNA strand. This approach boasts rapidity, being fast, because of stems from its automated loading of fluorescent ddNTPs and the integration of interconnected system components such as personal computers [4]. 

And the reference has been changed: Gupta, N.; Verma, V.K. Next-Generation Sequencing and Its Application: Empowering in Public Health Beyond Reality. Microbial Technology for the Welfare of Society 2019, 17, 313, doi:10.1007/978-981-13-8844-6_15.

Please I replied in detail to be as clear as possible, but these details are not all necessary in the manuscript However the sentence in lines 48-50 must be rephrased and reference n.3 is incorrect and needs to be changed: the first modified systems were released in the ‘90s. -line 48-50 has been rephrased and the reference has been changed:

Second big step in developing molecular diagnostic was first generation sequencing. Sanger introduced his "plus and minus" method for DNA sequencing in 1970s. This pioneering technique marked a crucial transition, paving the way for the subsequent development of sequencing methods that have since become the dominant approach in the field for the past three decades[3].

[3] Sanger, F.; Coulson, A.R. A Rapid Method for Determining Sequences in DNA by Primed Synthesis with DNA Polymerase. J Mol Biol 1975, 94, doi:10.1016/0022-2836(75)90213-2.

New Line 252  (Old Line 226)

  1. Many diseases have a genetic basis and are inherited [57]. “ To support this sentence is cited a paper dated 2000 on a mailed questionnaire to survey the feelings of Finnish Dog-Owners about Programs to Control Canine 645 Genetic Diseases” (n.57: Leppänen, M.; Paloheimo, A.; Saloniemi, H. Attitudes of Finnish Dog-Owners about Programs to Control Canine 645 Genetic Diseases. Prev Vet Med 2000, 43, 145–158, doi:10.1016/S0167-5877(99)00098-7.) – A new citation has been added.

REV This added paper is more about genetic counseling (such as the previous n.57 ) and inbreeding control. They are neither centered nor meaningful to the topic. Here citation of OMIA (whose citation has already been suggested) includes everything. Please, exclude the two cited papers (or if you strongly need to leave them) but add: Nicholas, F.W., Tammen, I., & Sydney Informatics Hub. (1995). Online Mendelian Inheritance in Animals (OMIA) [dataset]. https://omia.org/. https://doi.org/10.25910/2AMR-PV70. The citation date is 1995, but the site is constantly updated (https://omia.org/home/)

Suggested citation has been added, previous has been deleted.

Reviewer 3 Report

Comments and Suggestions for Authors

The manuscript has thoroughly improved. I do not have further suggestions.

Author Response

Thank you for your valuable suggestions and your time.